# Preparation of Sesquiterpene Lactone-Loaded PLA Nanoparticles and Evaluation of Their Antitrypanosomal Activity

**DOI:** 10.3390/molecules24112110

**Published:** 2019-06-04

**Authors:** Njogu M. Kimani, Solveig Backhaus, Josphat C. Matasyoh, Marcel Kaiser, Fabian C. Herrmann, Thomas J. Schmidt, Klaus Langer

**Affiliations:** 1Institute of Pharmaceutical Biology and Phytochemistry (IPBP), University of Muenster, PharmaCampus Corrensstrasse 48, D-48149 Muenster, Germany; m_kima01@uni-muenster.de (N.M.K.); shermanic@uni-muenster.de (F.C.H.); 2Institute of Pharmaceutical Technology and Biopharmacy, University of Muenster, PharmaCampus Corrensstrasse 48, D-48149 Muenster, Germany; sbackhau@uni-muenster.de; 3Department of Chemistry, Egerton University, P.O. Box 536, Egerton 20115, Kenya; josphat2001@yahoo.com; 4Swiss Tropical and Public Health Institute (Swiss TPH), Socinstrasse 57, CH-4051 Basel, Switzerland; marcel.kaiser@unibas.ch; 5University of Basel, Petersplatz 1, CH-4003 Basel, Switzerland

**Keywords:** human African trypanosomiasis, sesquiterpene lactones, nanoparticles, PLA, *Trypanosoma brucei*

## Abstract

Human African trypanosomiasis (HAT), also commonly known as sleeping sickness, is a neglected tropical disease affecting millions of people in poorly developed regions in sub-Saharan Africa. There is no satisfactory treatment for this infection. The investment necessary to bring new drugs to the market is a big deterrent to drug development, considering that the affected communities form a non-lucrative sector. However, natural products and many sesquiterpene lactones (STLs) in particular are very strong trypanocides. Research and applications of nano-drug delivery systems such as nanoparticles (NPs) have undergone unprecedented growth in the recent past. This is mainly due to the advantages offered by these systems, such as targeted delivery of the drug to the place of action (cell, parasite, etc), sustained release of the drug, increased bioavailability, reduced drug dosage, and reduction of undesired side effects, among others. In this study, the STLs α-santonin, arglabin, schkuhrin II, vernolepin, and eucannabinolide, all trypanocides, were loaded into polylactic acid (PLA) NPs through an emulsification-diffusion method. The NPs were stable, homogenous, and spherical in shape with a rounded knotty depression like a navel orange. The average particle sizes were 202.3, 220.3, 219.5, 216.9, and 226.4 nm for α-santonin, arglabin, schkuhrin II, vernolepin, and eucannabinolide, respectively. The NPs had encapsulation efficiencies of 94.6, 78.1, 76.8, 60.7, and 78.9% for α-santonin, arglabin, schkuhrin II, vernolepin, and eucannabinolide, respectively. The NPs loaded with arglabin, vernolepin, and eucannabinolide exhibited considerable antitrypanosomal activity against *Trypanosoma brucei rhodesiense (Tbr)* with free drug equivalent IC_50_ values of 3.67, 1.11 and 3.32 µM, respectively. None of the NP formulations displayed cytotoxicity towards mammalian cells (rat skeletal myoblast cell line L6). These results provide new insights into the possibility of incorporating STLs into nanoparticles, which may provide new options for their formulation in order to develop new drugs against HAT.

## 1. Introduction

Human African trypanosomiasis (HAT) is a fatal vector-borne parasitic neglected tropical disease affecting millions of people in poorly developed regions of sub-Saharan Africa. The disease is caused by two subspecies of a protozoan parasite, namely, *Trypanosoma brucei rhodesiense (Tbr)* and *Trypanosoma brucei gambiense*, which are transmitted to humans by the tsetse fly vector of the genus *Glossina*, endemic to sub-Saharan Africa, during blood-feeding [1]. *Trypanosoma brucei gambiense* causes a slow-progressing form of HAT in 24 countries of western and central Africa, while *Tbr* causes a faster-progressing form in 13 countries in eastern and southern Africa [1]. *Trypanosoma brucei gambiense* is responsible for over 98% of all HAT reported cases, while *Tbr* accounts for less than 2% [1,2]. The communities affected by these infections are poor and therefore do not offer a generation of revenue incentive to pharmaceutical companies for serious drug development. However, a few private organizations such as the Drugs for Neglected Diseases initiative (DNDi) invest in the search of new drugs. As a consequence of the long-term neglect, only a few chemotherapeutic options are available for this infection, and these involve high toxicity, high cost, difficulty in administration, and unavailability for resource-deprived rural communities [3]. In spite of recent new developments such as fexinidazole as the first oral treatment for HAT [4], the search for new efficacious and safe treatment agents with possibly new mechanisms of action is still of high importance. A variety of STLs have been shown to be highly potent against *T. brucei* parasites in several in vitro studies [5,6,7,8].

Nanotechnology has evolved greatly and gained exceedingly wide applications. In medicine, nanotechnology is of great interest due to the advantages and applicability offered by nanoformulations such as enhanced bioavailability, reduction in toxicity, improved solubility, reduced dosage, targeting by use of cell-specific ligands, improved drug efficiency by overcoming resistance mechanism, and increased half-life of drugs. One of the exploited formulations in nanotechnology is nanoparticles (NPs), particles ranging from 1 to 1000 nm [9,10], because of their unique physical, chemical, and biological properties. These unique properties are due to their relatively large surface area to volume ratio, stability in chemical processes, high mechanical strength, and increased reactivity [10]. Poly(lactic acid) (PLA), a biodegradable and biocompatible thermoplastic aliphatic polyester obtained from lactic acid monomers, is one of the polymers used in the formulation of NPs. It has attained the Generally Recognized as Safe (GRAS) status of the Food and Drug Administration (FDA) and has been extensively studied as a drug carrier [11,12]. Polymeric NPs have been employed successfully as a drug delivery system in many instances and with great potential in antitumor, antiviral, antifungal, and genetic therapy as well as antitrypanosomal treatment, among others [13,14]. In the latter case, pentamidine-loaded PEGylated poly(d,l-lactide-co-glycolide) (PLGA) NPs coupled to a single domain heavy chain antibody fragment (nanobody) that specifically recognizes the surface of the protozoan pathogen *T. brucei* were formulated by Arias et al. [15] The NPs showed a 7-fold decrease in IC_50_ value of the formulation relative to free pentamidine in vitro. Additionally, in vivo studies using a murine model of African trypanosomiasis demonstrated that the formulation cured all infected mice at a 10-fold lower dose than the minimal full curative dose of free pentamidine and 60% of mice at a 100-fold lower dose [15]. Similarly, pentamidine-loaded PEGylated-chitosan NPs coupled to the same single domain nanobody were found to circumvent drug resistance in a resistant cell line as a result of mutations in the surface transporter that mediate drug uptake. This was because pentamidine-loaded NPs enter trypanosomes through endocytosis instead of via classical cell surface transporters. Furthermore, the NPs were able to reduce minimal curative dose [16].

Thus, the aim of the present study was to develop a polymeric nanoparticulate system for STLs and to demonstrate the antitrypanosomal efficacy of the resulting formulation. α-Santonin, arglabin, schkuhrin II, vernolepin, and eucannabinolide, STLs of different structural subtypes that had shown different levels of antitrypanosomal activity in previous investigations [6,8,17,18], were included in this study. The NPs were characterized with respect to surface charge, particle diameter, encapsulation efficiency, and drug loading. Most importantly, the antitrypanosomal activity of the developed NPs against bloodstream forms of *Tbr* was investigated and their selectivity for the parasite assessed by comparison with their cytotoxicity against mammalian cells, specifically the rat skeletal myoblast cell line L6, was determined.

## 2. Results and Discussion

In the present study, sesquiterpene lactone-loaded PLA-NPs were prepared by an emulsion-diffusion technique. The choice of PLA as material for the NPs was made because this polymer is biocompatible, biodegradable, and FDA-approved [19,20]. It is also approved for use in hygiene products, as a wrapping and catering material, for agricultural and medical technology uses, and in sportswear, among other uses [12,21]. Table 1 summarizes the size, polydispersity indices, zeta potential, encapsulation efficiency, and drug load. The percentage yield (recovery) for each set of NPs, calculated from the evaluation of the mass of NP recovered and the initial mass of polymer and drug used for the preparation of NPs, was between 44% and 57%. The mass of stabilizer (in this case poly(vinyl alcohol) (PVA)) was not used to calculate the percentage yield in this case however, the variability and quantity of the yield were largely influenced by the method of purification and recovery which involved repetitive centrifugation and resuspension steps. The polydispersity indices were all between 0.02 and 0.10, indicating almost monodisperse particle size distributions (Table 1).

The surface morphology of the NPs was examined by atomic force microscopy (AFM) as shown in Figure 1. Generally, all the NPs were homogenous and spherical in shape with a rounded knotty depression like a navel orange. The average particle sizes were 202.3, 220.3, 219.5, 216.9, and 226.4 nm for α-santonin, arglabin, schkuhrin II, vernolepin, and eucannabinolide, respectively. For all of the preparations the particle diameter was solely defined by the used emulsion-diffusion technique with no significant differences in the particle diameter of the different formulations (ANOVA). Usually, the NPs are expected to be matrix-type drug delivery system, with the drug dispersed within the polymer matrix, because the NPs are formed from an emulsion system in which both the polymer and the drug are co-dissolved in the inner organic phase before NP formation.

The stability of the NPs was evaluated by measuring the zeta potential of the colloidal system. The zeta potential in all cases was between −26 and −36 mV, indicating a sufficient electrostatic stabilization of the NPs, and hence did not tend to aggregate. Additionally, the NPs were sterically stabilized by the used PVA. These, highly charged, NPs are stable as colloidal suspensions since the Coulombic repulsion forces arising from the surface charge can overcome the van der Waals attractive forces between them [22]. The zeta potential of the NPs formulated in this study is expected to be dependent on the carboxylic PLA end groups, the PVA (stabilizer) and the encapsulated STL. However, other than α-santonin, the other STLs do not seem to have an effect on the zeta potential based on the comparable zeta potential values (Table 1) of the empty and STL-loaded NPs.

### 2.1. Drug Loading and Encapsulation Efficiency

In the development of new drug delivery systems such as nanoparticles, the drug load is an important physicochemical parameter that should be carefully determined. The HPLC methods developed herein (see Appendix A for more details) were applied to accurately determine the STLs encapsulated into PLA-NPs. The encapsulation efficiency accounts for the drug loss in the nanoparticle preparation process and the drug load explains the amount of the drug in the nanoparticle. The encapsulation efficiency was determined by the indirect method based on the quantification of the non-encapsulated drug in the supernatants. As can be seen from Table 1, α-santonin and arglabin showed the best loading efficiencies. The encapsulation efficiencies were 94.6, 78.1, 76.8, 60.7, and 78.9% for α-santonin, arglabin, schkuhrin II, vernolepin, and eucannabinolide, respectively (Table 1). Formulation of α-santonin PLA-NP had the highest encapsulation efficiency. The encapsulation efficiency in decreasing order was α-santonin > eucannabinolide > arglabin > schkuhrin II > vernolepin. The encapsulation efficiency was related to the aqueous solubility and lipophilicity of the STLs. Vernolepin has the highest calculated logS at −1.88 (Appendix A) and is the least lipophilic (logP = 1.06) while α-santonin is the most lipophilic (logP = 2.60) and and least soluble in aqueous solution (logS = −3.45). Hence, the encapsulation was highest in α-santonin and lowest in vernolepin as expected because the emulsification-diffusion method employed in this formulation is best suited for encapsulation of lipophilic drugs into nanospheres [23]. The drug load was highest in α-santonin NPs with 42.6%, whereas arglabin, schkuhrin II, eucannabinolide, and vernolepin NPs showed drug loads of 7.5, 2.5, 2.5, and 0.5% respectively.

### 2.2. Drug Release Profile

In vitro release of STLs from lyophilized STL-PLA NPs was evaluated at a physiological pH by incubating 1 mg NPs in 1 mL phosphate buffered saline (pH = 7.5) at 37 °C. A quick release of the drug followed by a slower exponential stage was observed in all cases (Figure 2). About three-quarters of the initially incorporated STLs was detected in the supernatant of the redispersed nanoparticles after approximately 6 h for α-santonin, arglabin, vernolepin, and eucannabinolide. However, for schkuhrin II, this was achieved after 36 h. The slow release of schkuhrin II is not clearly understood. However, schkuhrin II had the highest topological polar surface area (TPSA = 139.59 Å^2^). Eucannabinolide, with the second highest TPSA (119.36 Å^2^; Appendix A) had an initial slow release. Further analysis of drug release for these formulations is therefore warranted.

### 2.3. Bioactivity of Formulated Nanoparticles

Drug delivery systems may provide selective delivery of drugs, antitrypanosomal agents in this case, to the target cells and therefore improve antiparasitic chemotherapy. These systems must, therefore, increase the therapeutic index of the drug by decreasing its toxicity without loss of efficacy. In this study, the loaded NPs were tested in vitro against *Tbr* (STIB 900 strain) trypomastigotes and for cytotoxicity against mammalian cells (rat-skeletal myoblast cell line L6). The results are summarized in Table 2.

α-Santonin loaded NPs had IC_50_ values of >50 µg/mL for both *Tbr* and cytotoxicity (see Table 2). In contrast, α-santonin has been reported to have IC_50_ values of 234.5 and >400 µg/mL for *Tbr* and cytotoxicity, respectively [17]. The low activity of the α-santonin-PLA-NPs is thus consistent with the low activity of free α-santonin. It should be noted that α-santonin was included in this study, not because of its bioactivity but due to its commercial availability at a low price as a model STL for the purpose of investigating the general possibility to encapsulate STLs in PLA-NPs.

Arglabin-loaded NPs displayed an IC_50_ value of 12.15 ± 3.68 µg/mL against *Tbr* and a cytotoxicity IC_50_ value of 40.33 ± 10.46 µg/mL. This corresponds to free drug equivalent IC_50_ values of 3.67 ± 0.28 and 12.28 ± 0.78 µM for *Tbr* and cytotoxicity, respectively. These IC_50_ values were higher than those determined for the unencapsulated STL i.e., 2.52 ± 0.42 and 6.18 ± 0.28 µM for *Tbr* and cytotoxicity, respectively [17] (see Table 2). Vernolepin loaded NPs showed IC_50_ values of 61.3 ± 3.65 and >100 µg/mL against *Tbr* and cytotoxicity, respectively. The anti-*Tbr* activity of vernolepin loaded NPs’ free drug IC_50_ value equivalence was 1.11 ± 0.02 µM, which was also higher than that of unformulated drug (0.19 ± 0.04 µM [8]) (see Table 2). However, these NPs showed very low cytotoxicity values and can thus be considered for further evaluation. Similarly, eucannabinolide-loaded NPs had an anti-*Tbr* IC_50_ value of 55.8 ± 4.68 µg/mL which is equivalent to 3.32 ± 0.12 µM of the STL contained in these NPs. This is, in turn, higher than the IC_50_ value of 1.14 ± 0.08 µM [6] of the unformulated STL (see Table 2). The eucannabinolide loaded NPs also displayed a low cytotoxicity value with an IC_50_ value of >100 µg/mL and thus can be considered non-cytotoxic. In case of schkuhrin II, the STL loaded NPs had IC_50_ values of >100 µg/mL for both *Tbr* and cytotoxicity, i.e., were essentially inactive in spite of the relatively high antitrypanosomal activity of the free STL. This could be attributed to the slow release associated with this formulation and the high TPSA of schkuhrin II.

These results indicate that PLA-NPs are potential carriers of STLs despite the fact that free STLs have higher antitrypanosomal activity than STL-loaded NPs. It has to be kept in mind that drug-loaded NPs often have superior properties to free drug such as improved pharmacokinetics, prolonged and controlled release of the drug [23,24]. Moreover, the low cytotoxic activity of the NPs is a clear indicator that more studies on the applicability of STL-PLA-NPs may be useful and that the potential of encapsulated STLs should be explored further.

## 3. Materials and Methods 

### 3.1. Materials

Poly(d,l-lactide) (PLA, Resomer^®^ R 203H) was obtained from Evonik Industries AG (Essen, Germany); poly(vinyl alcohol) (PVA, average molecular weight 31 kDa, 85–89% hydrolyzed) and trehalose were purchased from Merck KGaA (Darmstadt, Germany). The sesquiterpene lactones (STLs) were from the following sources: α-santonin (**1**) was obtained from Karl Roth GmbH & Co. (Karlsruhe, Germany), arglabin (**2**) was isolated from *Artemisia glabella* (Asteraceae) [17,18], vernolepin (**3**) from *Vernonia lasiopus* (Asteraceae) [8], and eucannabinolide (**4**) as well as schkuhrin II (**5**) from *Schkuhria pinnata* (Asteraceae) [6] (Figure 3).

### 3.2. Preparation of PLA Nanoparticles

The STL-containing polylactic acid (PLA) nanoparticles (NPs) were prepared by an emulsification-diffusion method. In brief, 10 mg of STL and 100 mg of PLA were dissolved in 2 mL CH_2_Cl_2_ to constitute an organic phase. The organic phase was added to 6 mL aqueous solution of PVA (2%, *w*/*v*). The resulting mixture was homogenized by a high shear mixer (Ultra Turrax, IKA, Staufen, Germany) in an ice bath for 30 min at 24,000 rpm and diluted with 6 mL PVA (1%, *w*/*v*) solution. The CH_2_Cl_2_ was removed by stirring (500 rpm) the emulsion overnight at room temperature. Finally, the particles were collected by centrifugation at 20,000 g for 15 min and washed twice with purified water. For the empty NPs, the process was performed without STL.

The NPs were then lyophilized using an Epsilon 2–4 freeze-dryer (Martin Christ Gefriertrocknungsanlagen GmbH, Osterode am Harz, Germany). Briefly, aliquots of 100 μL NPs suspension were added to 100 μL trehalose solution (6% *w*/*v*) as a cryoprotective agent in 2 mL lyovials. Then, samples were frozen at −40 °C for 3 h followed by a primary drying step at 0.05 mbar and −34 °C for 24 h. For secondary drying, the temperature was then raised to 20 °C for 11 h while applying a vacuum of 0.025 mbar. After drying, the vials were sealed and stored at 4 °C until further use.

### 3.3. Characterization of Nanoparticles

The NPs’ particle diameter and polydispersity index were determined by photon correlation spectroscopy (PCS) and zeta potential was measured by microelectrophoresis using a Malvern Zetasizer Nano ZS (Malvern Instruments GmbH, Herrenberg, Germany). For particle size determination the NP suspension was diluted with ultrapure water (1:100) in a disposable cuvette right before use and measured at a temperature of 22 °C using a backscattering angle of 173°. The zeta potential was measured in the same instrument by laser Doppler microelectrophoresis to provide information about the surface charge of the NP. Therefore, the NP dilutions described above were transferred to a folded capillary cell and the determination was conducted at 22 °C.

### 3.4. Morphology of Nanoparticles

The morphological analysis of nanoparticles was performed on a Bruker Dimension 3100 atomic force microscope (AFM), equipped with a Nanoscope IIIa controller (Bruker, Karlsruhe, Germany). Before the analysis, 5 µL of NPs suspension (0.25 mg/mL) was transferred onto a clean glass slide and air dried. Images of NPs were then obtained in intermittent contact mode (Tapping Mode) with n-type silicon cantilevers (HQ:NSC14/Al BS, nominal tip radius <10 nm, typical resonance frequency of about 160 kHz, nominal spring constant of 5 N/m; manufactured by µmash, Sofia, Bulgaria) 2% below resonance frequency at a root mean square (RMS) amplitude of around 2 Volts. Further data analysis was carried out using NanoScope Analysis version 1.5 (Bruker, Billerica, MA, USA).

### 3.5. Quantification of Encapsulated STLs by HPLC

The amount of STL incorporated into the NPs was determined indirectly by validated HPLC methods. Briefly, an aliquot of the supernatant (100 µL) obtained after purification of the NPs was diluted with 1.9 mL acetonitrile (MeCN). The solution was centrifuged for 30 min at 20,000 g and the supernatant filtered through a microfilter. The chromatographic separation was carried out using aliquots (10 μL) of the diluted supernatant. The HPLC methods were validated in terms of specificity, linearity, precision, accuracy, stability, limit of detection (LOD), and limit of quantification (LOQ) according to International Council for Harmonisation (ICH) guideline [25] (for more details see Appendix A).

### 3.6. Drug Release Profile

To determine in vitro release of STLs from the NPs, 1 mg NPs were incubated in 1 mL phosphate buffer (0.1 mM, pH = 7.5) at 37 °C in a thermal shaker. Samples were individually prepared for each of the following points in time: 0, 0.5, 1, 3, 5, 7, 24, 36, and 48 h. The samples were centrifuged at 20,000 g for ten minutes and the amount of released STL in the supernatant determined by HPLC as described above.

### 3.7. Bioactivity Assays

#### 3.7.1. Trypanosoma Brucei Rhodesiense

Minimum essential medium with Earle’s salts (50 µL) supplemented with 0.2 mM 2-mercaptoethanol, 1 mM Na-pyruvate and 15% heat-inactivated horse serum was added to each well of a 96-well microtiter plate. Serial NPs dilutions covering a range from 100 to 0.002 µg/mL were prepared. Then, 1 × 10^4^ bloodstream forms of *Tbr* STIB 900 in 50 µL of the medium were added to each well and the plate was incubated at 37 °C under a 5% CO_2_ atmosphere for 72 h. Ten µL of resazurin solution (12.5 mg resazurin dissolved in 100 mL distilled water) were then added to each well and incubation continued for a further 2–4 h [26]. The plate was then read in a Spectramax Gemini XS microplate fluorometer (Molecular Devices Corporation, San Jose, CA, USA) using excitation and emission wavelengths of 536 nm and 588 nm, respectively. Fluorescence development was measured and expressed as a percentage of the control. The IC_50_ values were calculated from the obtained data having been transferred into the graphic program Softmax Pro (Molecular Devices, Corporation, USA). Melarsoprol was used as a positive control.

#### 3.7.2. Cytotoxicity

Cytotoxicity was determined using rat skeletal myoblasts (L6 cells; CRL1458 obtained from American Type Culture Collection ATCC, Manassas, VA, USA) in a similar protocol as described for *Tbr*. L6 cells were seeded into RPMI 1640 medium supplemented with L-glutamine 2 mM, HEPES 5.95 g/L, NaHCO_3_ 2 g/L and 10% fetal bovine serum in 96-well microtiter plates (4,000 cells/well). Serial NP dilutions covering a range from 100 to 0.002 µg/mL were prepared and the plate was incubated at 37 °C under a 5% CO_2_ atmosphere for 72 h. Ten µL of resazurin solution (12.5 mg resazurin dissolved in 100 mL distilled water) were then added to each well and incubation continued for a further 2 h. The plate was then read in a Spectramax Gemini XS microplate fluorometer (Molecular Devices Corporation, San Jose, CA, USA) using excitation and emission wavelengths of 536 nm and 588 nm, respectively [26]. Fluorescence development was measured and expressed as a percentage of the control. Subsequently, IC_50_ values were calculated from the sigmoidal inhibition curves. Podophyllotoxin was used as a positive control. 

## Figures and Tables

**Figure 1 molecules-24-02110-f001:**
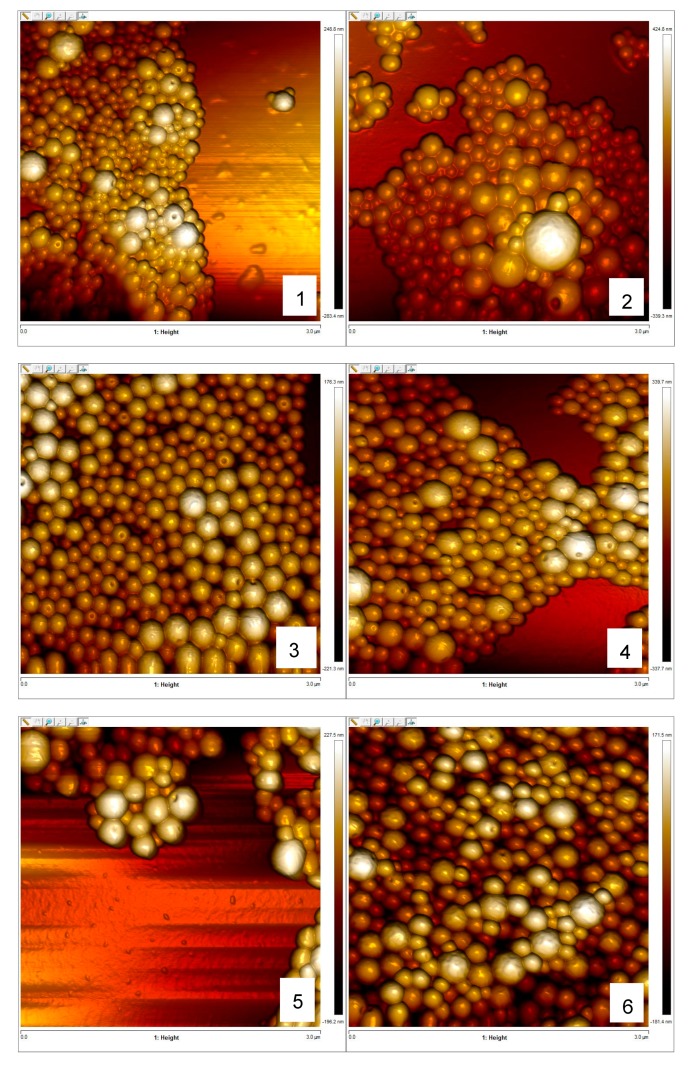
Atomic force microscopy images of formulated nanoparticles. (**1**) Empty polylactic acid nanoparticles (PLA-NPs), (**2**) α-santonin-PLA-NPs, (**3**) arglabin-PLA-NPs, (**4**) schkuhrin II-PLA-NPs, (**5**) vernolepin-PLA-NPs, and (**6**) eucannabinolide-PLA-NPs.

**Figure 2 molecules-24-02110-f002:**
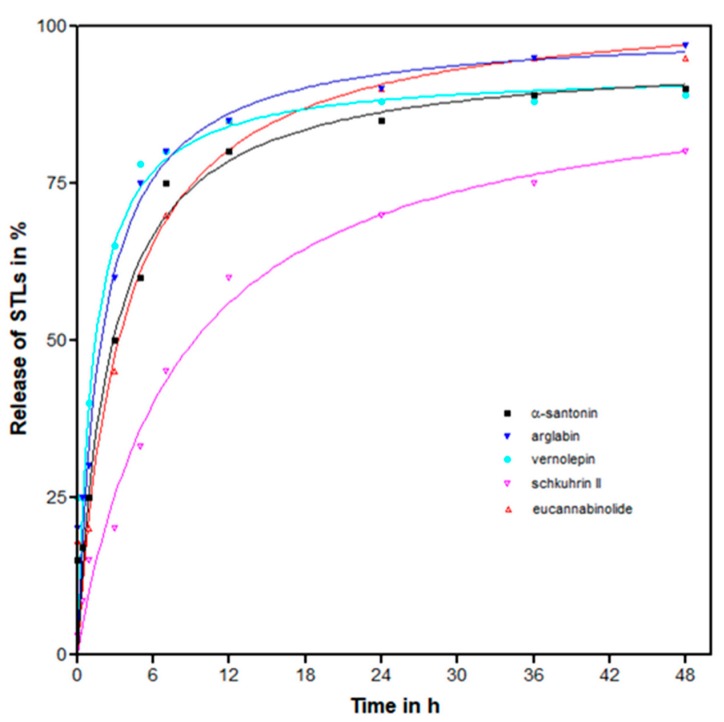
Sesquiterpene lactones (STLs) released over time from STL-PLA-NPs. Trendlines are polynomial (2nd order) with R^2^ values ≥0.90 in all cases.

**Figure 3 molecules-24-02110-f003:**
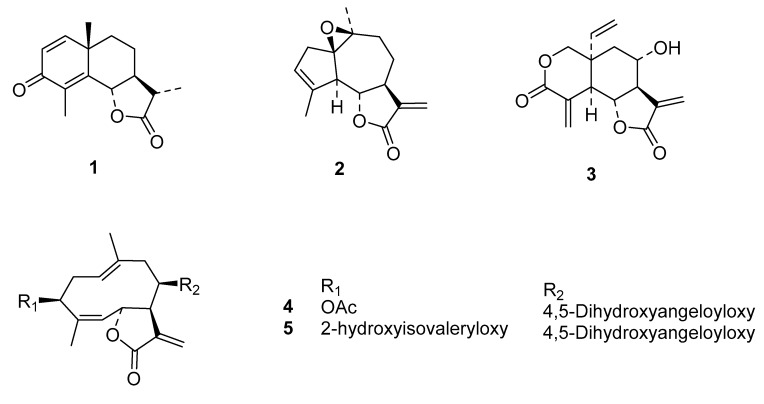
Sesquiterpene lactones used in this study.

**Table 1 molecules-24-02110-t001:** Physicochemical characteristics of sesquiterpene lactone loaded polylactic acid nanoparticles (PLA-NPs) (mean ± S.D., *n* = 3).

Formulation	Particle Diameter (nm)	Polydispersity Index (PDI)	Zeta Potential (mV)	Encapsulation Efficiency (%)	Drug Load (%)
PLA-NP	208.9 ± 10.5	0.05 ± 0.02	−36.1± 5.4	-	-
α-Santonin	202.3 ± 8.2	0.03 ± 0.01	−26.3 ± 7.8	94.6 ± 2.2	42.6 ± 8.2
Arglabin	220.3 ± 12.8	0.02 ± 0.00	−35.3 ± 5.6	78.1 ± 7.4	7.5 ± 1.3
Schkuhrin II	219.5 ± 9.9	0.05 ± 0.01	−35.4 ± 4.9	76.8 ± 3.9	2.5 ± 0.2
Vernolepin	216.9 ± 16.1	0.10 ± 0.01	−35.3 ± 6.7	60.7 ± 8.9	0.5 ± 0.3
Eucannabinolide	226.4 ± 10.2	0.02 ± 0.00	−33.5 ± 5.3	78.9 ± 6.3	2.5 ± 0.7

**Table 2 molecules-24-02110-t002:** Bioactivity data (IC_50_ values) of the STL-loaded NPs, their equivalent free drug activity, and activity of naked STLs (mean ± S.D., *n* = 3 unless otherwise stated).

	STL Loaded NPs *Tbr* (µg/mL)	Equivalent Free STL *Tbr* (µM)	Free STL *Tbr* (µM)	NPs Cytotoxicity (µg/mL)	Free STL Cytotoxicity (µg/mL)
α-Santonin	>50 ^c^		234.50 ^a^	>50 ^c^	>50 ^c^
Arglabin	12.15 ± 3.68	3.67 ± 0.28	2.52 ± 0.42 ^b^	40.33 ± 10.46	1.52 ± 0.68
Schkuhrin II	>100 ^c^		0.82 ± 0.17 ^b^	>100 ^c^	5.24 ± 0.56
Vernolepin	61.30 ± 3.65	1.11 ± 0.02	0.19 ± 0.04 ^b^	>100 ^c^	0.74 ± 0.05
Eucannabinolide	55.80 ± 4.68	3.32 ± 0.12	1.14 ± 0.08 ^b^	>100 ^c^	3.28 ± 0.83

^a^ Only one replicate used to determine IC_50_; ^b^ Data are means of two independent determinations ± absolute deviation; ^c^ Highest concentration tested; IC_50_ not determined. *Tbr*: *Trypanosoma brucei rhodesiense.*

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
