# Peer review of "Preparation of Sesquiterpene Lactone-Loaded PLA Nanoparticles and Evaluation of Their Antitrypanosomal Activity"

_molecules, 2019, doi:10.3390/molecules24112110_

Round 1

Reviewer 1 Report

Good work. Please explain the self-contradictory statements in the manuscript.

Details in attachments.

Author Response

Reviewer 1

1. Particle size of blank nanoparticles was around 208 nm and that of α-santonin was around 202 nm. %Drug loading is around 42%, still the particle size is smaller than the blank NPs. Explain?

Answer: The particle diameter of all formulations was solely caused by the emulsion-diffusion technique used for particle preparation with no statistical significant difference between the different formulations (ANOVA). This was clarified on page 3, lines 118f. The comparatively high drug load of santonin-loaded nanoparticles is probably due to a low yield of PLA-NPs.

2. Zeta potential of most formulations is around -36mV. A zeta potential less tha -30mV suggests instability of formulation and aggregation of particles. Point mentioned in manuscript itself 123-124. Contradiction in results and facts stated.

Answer: The Zeta potential of the NPs in all cases was between -26 and -36 mV. These  values are within the acceptable range and hence the NPs are quite stable. Additionally the NPs were sterically stabilized by the used poly(vinyl alcohol) which is well known to form a protective polymer layer on the surface of the NP.

3. Table 1 indicates monodisperse formulations however high polydispersity can be seen from AFM data especially in case of α-santonin NPs.

Answer:  DLS and AFM are complementary techniques in characterization of NPs however necessary care should be taken in the interpretation of data from the two techniques. The difference in PI observed between the two methods could not be explained but further investigations are being considered in future studies.

4. Check the scale for AFM images 3 and 6.

Answer: It is not clear what the reviewer means. All scales are identical as shown in the figures.

5. Table 1 indicates, % drug loading for vernolepin NPs was 0.5% while text says it was 2.5% (Line 157). When entrapment efficiency is 60% drug loading is only 0.5%, explain the suitability and efficiency of the method.

Answer: Corrected. The objective of this work was to encapsulate several STLs into PLA NPs and the method was efficient to varying degrees depending on the STL in question. Moreover, the vernolepin formulation was active against Tbr and was also not cytotoxic.

6. Bioactivity and cytotoxicity data is not clear. Knowing the %drug loading the IC50 values should have been indicated in μM or in μg/ml for clear comparison. No data is provided for the effect of blank nanoparticles on the parasite and L6 cell line.

Answer: The IC50 values are indicated in µM in Table 2 for clear comparison with the unencapsulated STL.

7. On what basis dose of sesquiterpenes was finalized i.e. 1:10 ratio with PLA?

Answer: This ratio was selected arbitrarily.

8. How the complete removal of CH2Cl2 was ensured and what grade of the solvent was used?

Answer: The dichloromethane was removed completely by stirring the emulsions overnight. The Solvent was of analytical grade.

9. The objective of study was to combine the benefits of nanotechnology with STLs, but bioactivity studies indicate free STLs have better results. The authors claim that nanotechnology has other advantages even though NPs led to lowering of antitrypanosomal activity, so either the importance of this study should be explained or further studies should be carried out in support of the statement 208-213.

Answer: The advantage of the nanoparticles is that they are not cytotoxic whereas the pure STLs are. The cytotoxicity is reduced to a larger extent than the antitrypanosomal activity. This is already stated in the original manuscript. However, we added the cytotoxic activity data of the free STLs to the table and thus made an even clearer statement.

10. Also carrying out cytotoxciity studies on one type of cell line is not enough. More cell lines should be incorporated of human origin.

Answer: The scope of this study is not to investigate the cytotoxic activity against all possible cells but to obtain data for comparison with those previously obtained. The cytotoxicity of the nanoparticles has therefore been tested against the same cell line as that used previously for the pure STLs. No further tests appear necessary for this manuscript.

11. As the emphasis has also laid on environmentally benign nature PLA polymer. Please include other polymeric nanoparticles ( cellulose, lignin and others) suggested for drug delivery. These

J. Bhandari et al., Int J Nanomedicine. 12, 2021–2031 (2017).,P. K. Mishra, A. Ekielski, Nanomaterials. 9, 243 (2019). papers can be used.

Answer: It is not the scope of the present study to investigate all kinds of different materials. PLA was chosen here because it is frequently used and due to its chemical properties could be expected to incorporate the compounds under study. Further steps in the direction of using other materials might be taken in further studies.

We thank this reviewer for the constructive and very thorough cricitism and hope that our response will convince her or him of our work.

Reviewer 2 Report

The manuscript submitted by is of very high interest for Molecules, the manuscript is well written and data support the conclusions. The manuscript can be published with modifications listed below:

The introduction could be revised by focussing  on antiparasitic properties of nanoparticles.

see for example: Mesoporous silica nanocarriers encapsulated antimalarials with high therapeutic performance Saliu Alao Amolegbe, Yui Hirano, Joseph Oluwatope Adebayo, Olusegun George Ademowo, Elizabeth Abidemi Balogun, Joshua Ayoola Obaleye, Antoniana Ursine Krettli, Chengzhong Yu &Shinya Hayami.

Scientific Reports, volume 8, Article number: 3078 (2018).

Typographical error:

Synthesis of PLA, replace ml by mL.

Author Response

Reviewer 2

The manuscript submitted by is of very high interest for Molecules, the manuscript is well written and data support the conclusions. The manuscript can be published with modifications listed below:

The introduction could be revised by focussing  on antiparasitic properties of nanoparticles.

see for example: Mesoporous silica nanocarriers encapsulated antimalarials with high therapeutic performance Saliu Alao Amolegbe, Yui Hirano, Joseph Oluwatope Adebayo, Olusegun George Ademowo, Elizabeth Abidemi Balogun, Joshua Ayoola Obaleye, Antoniana Ursine Krettli, Chengzhong Yu &Shinya Hayami. Scientific Reports, volume 8, Article number: 3078 (2018).

Answer: The introduction is already focused on antiparasitic activity and utility of nanoparticles against Trypanosomes. The authors do not feel the need the widen this to antimalarial potential of nanoparticles since malaria (caused by a completely different parasite) is not under study here. We therefore did not include this further reference.

Typographical error:

Synthesis of PLA, replace ml by mL.

Answer: This was corrected.

We thank this reviewer for the constructive cricitism and hope that our response will convince her or him of our work

Reviewer 3 Report

The authors formulated PLA nanoparticles loaded with different subtypes of sesquiterpene lactones. These nanoparticles were well-characterised using with different methods: photon correlation spectroscopy, Malvern zeta sizer, atomic force microscope and HPLC. The antitrypanosomal activity of nanoparticles were also tested using different bioactivity assays: trypanosoma brucei rhodesiense test and cytotoxicity test on rat skeletal myoblasts. The drug release profile was measured by HPLC.

The methods are well-described. The experimental design is appropriate.

The results and the discussion  parts are well-written.

Author Response

Reviewer 3

The authors formulated PLA nanoparticles loaded with different subtypes of sesquiterpene lactones. These nanoparticles were well-characterised using with different methods: photon correlation spectroscopy, Malvern zeta sizer, atomic force microscope and HPLC. The antitrypanosomal activity of nanoparticles were also tested using different bioactivity assays: trypanosoma brucei rhodesiense test and cytotoxicity test on rat skeletal myoblasts. The drug release profile was measured by HPLC.

The methods are well-described. The experimental design is appropriate.

The results and the discussion  parts are well-written.

Answer:

We thank this reviewer for the positive evaluation.

Round 2

Reviewer 1 Report

Authors have addressed reviewer's concerns.